# Assessment of Volatiles and Polyphenol Content, Physicochemical Parameters and Antioxidant Activity in Beers with Dotted Hawthorn (*Crataegus punctata*)

**DOI:** 10.3390/foods9060775

**Published:** 2020-06-11

**Authors:** Alan Gasiński, Joanna Kawa-Rygielska, Antoni Szumny, Justyna Gąsior, Adam Głowacki

**Affiliations:** 1Department of Fermentation and Cereals Technology, Faculty of Biotechnology and Food Science, Wrocław University of Environmental and Life Science, Chełmońskiego 37 Street, 51-630 Wroclaw, Poland; joanna.kawa-rygielska@upwr.edu.pl (J.K.-R.); justyna.gasior@upwr.edu.pl (J.G.); adamglowackiag@gmail.com (A.G.); 2Department of Chemistry, Faculty of Biotechnology and Food Science, Wrocław University of Environmental and Life Sciences, C.K. Norwida street 25, 50-375 Wrocław, Poland; antoni.szumny@upwr.edu.pl

**Keywords:** antioxidant activity, *Crataegus punctata*, beer, total polyphenol content, gas-chromatography, headspace solid-phase microextraction, volatile compounds, aroma, sensory analysis

## Abstract

Beer with the addition of dotted hawthorn (*Crataegus punctata)* fruit and juice was prepared and analysed. The content of carbohydrates, glycerol and ethanol in beers was determined by high-performance liquid chromatography (HPLC). Analysis of the total content of polyphenols was also performed using the Folin-Ciocalteu method, as well as determining antioxidant capacity by DPPH^•^ and ABTS^+•^ assay, and the ability to reduce iron ions by FRAP assay. Content of volatile compounds was analysed by means of solid-phase microextraction and gas chromatography coupled with mass spectroscopy. Beers with addition of hawthorn, both juice and fruit, had higher antioxidative potential and higher polyphenols concentration compared to control beer. The content of polyphenols in beers was in the range 200.5–410.0 mg GAE/L, and the antioxidant activity was in the range of 0.936–2.04 mmol TE/L (ABTS^+•^ assay), 0.352–2.175 mmol TE/L (DPPH^•^ assay) and 0.512–1.35 mmol TE/L (FRAP assay). A sensory evaluation of beers was also carried out. Beer with hawthorn fruit addition obtained the best scores in sensory analysis for criteria such as aroma, taste and overall quality. This beer had the highest content of volatile compounds (287.9 µg/100 mL of beer), while the control beer had lowest concentrations (35.9 µg/100 mL of beer).

## 1. Introduction

Beer polyphenols come from malt (70–80%) and hops (20-30%) [1]. Polyphenols and melanoidins affect the antioxidant activity of beer, as well as its sensory properties [2]. Antioxidant compounds play a role in reducing the amount of free radicals present in human body. The imbalance between antioxidants and pro-oxidants is called oxidative stress, which is a cause of many diseases. Exogenous antioxidants, introduced into the body with food, can scavenge free radicals, chelate metal ions, or inhibit pro-oxidative enzymes, protecting our body from the negative effects of oxidative stress [3].

Moderate consumption of beer has a positive effect on some biomarkers associated with human health [4,5]. Food rich in polyphenols can reduce the possibility of developing many civilization-related diseases such as arteriosclerosis, cancer and neurodegenerative diseases. This means that there is a public interest in products with high content of polyphenols [6,7]. Quifer-Rada et al. identified 47 phenolic compounds in beer, including simple phenolic acids, hydroxycinnamoylquinics, flavanols, flavonols, flavones, alkylmethoxyphenols, alpha-acids and iso-alpha-acids, hydroxyphenylacetic acids and prenylflavonoids [8]. Phenols and polyphenols can directly contribute to the characteristics of beer, mainly to body, haze, flavour, fullness and astringency [9]. However, volatile compounds play a major role in creating the taste and aroma [10]. Sensory active chemicals in beer belong to various classes, such as esters, carbonyl compounds, alcohols, fatty acids, volatile phenols, furanic compounds or terpenoids [11]. In traditional beer styles volatile compound content depends on yeast metabolism and type of malt and hops used [12,13]. The aroma profile of beer is one of the main factors evaluated by consumers, so the analysis of volatile compound content is an important part of testing product quality.

Research on the impact of brewing raw materials and the production process on the content of polyphenolic compounds, antioxidant activity and beer physiochemical properties have already been described [14,15]. Beer characteristics can be modulated by using unusual ingredients in brewing technology such as fruits. Consumer interest in fruit beers is increasing [16]. The fruits or fruit juices used as raw materials in beer production, apart from changes in taste and aroma, add another source of bioactive compounds to the product [17,18]. Studies have been conducted on beers with the addition of cherry, raspberry, peach, apricot, grape, plum, orange and apple, but also less popular fruit like persimone or Cornelian cherry [18,19,20]. Researchers have also evaluated beers with the addition of lemon, grapefruit, raspberry and cranberry juice available on the market [21]. The possibilities of using industrial waste products, e.g., orange peels and eggplant peels, in the production of beer have also been explored [18,22].

The dotted hawthorn (*Crataegus punctata*) is a shrub that grows naturally in North America [23]. Analytical studies have shown that hawthorn fruit contain flavonoids such as kaempferol, apigenin, quercitrin, rutin, hesperidin and arbutin; phenolic acids such as ursolic and isovanillic acid; anthocyanins such as cyanidine-3-rutinoside, cyanidine-3-galactoside and cyanidine-3-arabinoside [24,25]. No work has been found in which it was analysed how the addition of hawthorn fruit or juice to beer modifies the physicochemical parameters, the content of polyphenols, the antioxidant activity and the content of volatile compounds in beers. This article describes comprehensive research on the use of hawthorn in beer production. A distinctive aspect of this research is the comparison of whole fruit or juice application. The influence of the additive form (juice or fruit) on beer properties shaping the quality of the product was discussed. Therefore, analysis of impact of fruit addition on content of volatile compounds and sensory analysis is an interesting aspect of conducted study.

This study aimed to determine whether the addition of dotted hawthorn fruit or juice could influence the physicochemical properties, polyphenol content, antioxidant activity, volatile compounds concentration and results of sensory analysis.

## 2. Materials and Research Methods

### 2.1. Reagents and Standards

Reagents used in this study were 1,1-diphenyl-2-picrylhydrazil (DPPH^•^) radical, diammonium salt of 2,2′-azobis (3-ethylbenzothiazoline-6-sulfonate) (ABTS^+•^), 2,4,6-tripyridyl-S-thiazine was used (TPTZ), 20% aqueous sodium carbonate solution, Folin-Ciocalteu reagent, acetic acid, sulfuric acid, FeCl_3_, sodium acetate and diatomaceous earth. Internal standard was 2-undecanone (Sigma-Aldrich, Saint Louis, MI, USA) with a content of 100 mg of compound per 100 mL of distilled water.

### 2.2. Biological Material

*Saccharomyces cerevisiae var. diastaticus* with the trade name “3724 Belgian Saison” from Wyeast company (Hood River, OR, USA) were used for wort fermentation. Yeasts were bought in liquid starter form which contains 1.2 × 10^12^ cells/mL. The wort was inoculated in dose of 4.3 mL/L according to the producers’ recommendations. The yeast strain used was low flocculating and possess high degree of fermentation (76–80%), with an optimal fermentation temperature in the range of 21–35 °C.

### 2.3. Research Material

The research material was American Saison-style beer in three variants: without any additions (BC), enriched with the addition of hawthorn fruit (BF) and enriched with hawthorn fruit juice (BJ). The raw materials were Pilsen malt, wheat malt, Vienna malt, Caramel Pale malt, Light Caramel malt, acidified malt. All malts were acquired from the Viking Malt company (Strzegom, Poland). Amarillo and Calypso hops (USA) were purchased from Twój Browar company (Wrocław, Poland). The hawthorn fruit (*Crataegus punctata)* of the variety “Aurea” and juice pressed from this fruit were added in the production process. The juice was obtained from fruits by the use of a manual winepress.

### 2.4. Brewing Technology

Mashing was carried out under laboratory conditions with an infusion system in the following conditions: 64 °C for 60 min and 72 °C for 15 min. Next, the whole mash was heated to 78 °C to inactivate the malt enzymes, filtered and 25 L wort was obtained. The wort was boiled for 75 min with addition of hops in three doses: first, Calypso (15 g), 75 min of boiling; second, Amarillo (15 g), 30 min of boiling; third, Amarillo (15 g) and Calypso (10 g), 15 min of boiling. The hopped wort was cooled to 25 °C, filtered and aerated. The initial extract content was set at 12.5% *(w/w)* measured using Densito 30PX densimeter (Mettler Toledo, DC, USA). Fermentation was carried out in fermentation vessels at 25 °C. Hawthorn juice or fruits were added to beers at a dose of 10% (*w*/*v*) in the last 3 days of fermentation. The resulting beer was aged in bottles for 3 weeks.

### 2.5. Physicochemical Parameters

The pH value of beer was measured using Mettler-Toledo MP 220 (Greifensee, Switzerland). The extract content was measured at 20 °C with a Densito 30P density meter (Mettler-Toledo, Columbus, OH, USA). The measurements were performed in triplicate.

### 2.6. Determination of Carbohydrate Profile, Ethanol and Glycerol Content

The sugar profile and the content of ethanol and glycerol were examined by means of High-Performance Liquid Chromatography (HPLC) [26]. Degassed and centrifuged (5500 rpm, 10 min) beer samples were diluted two-fold (1:1) with ultrapure water and filtered through nylon syringe filters with 0.45 µm pore size for chromatographic vials. The samples were then analysed using a Prominence liquid chromatography system (Shimadzu Corp., Kyoto, Japan) equipped with a Rezed ROA-Organic Acid H + column (300 × 4.6 mm) from Phenomex (Torrance, CA, USA). The following measurement parameters were used: sample volume: 20 µL, separation temperature: 60 °C, mobile phase flow rate: 0.6 mL/min, mobile phase: 0.005 M H_2_SO_4_, detection temperature: 50 °C. The concentration of ethanol, glycerol, dextrins, maltose, glucose and maltotriose was based on five-point calibration curves using Chromax 10.0 software (Pol-Lab, Wilkowice, Poland). All measurements were performed in triplicate.

### 2.7. Analysis of Total Polyphenol Content and Antioxidative Activity

#### 2.7.1. Analysis of Total Polyphenol Content

The total polyphenol content of the beer was determined using the Folin-Ciocalteu (F-C) spectrophotometric method [27]. Degassed and centrifuged beer samples or fruit juice samples (0.1 mL) and 0.2 mL of F-C reagent were pipetted into cuvettes. After 3 min, 1 mL of a 20% aqueous solution of sodium carbonate (Na_2_CO_3_) and 2 mL of distilled water were added. The absorbance at 765 nm was measured after 1 h using Beckmann DU650 spectrophotometer (Brea, CA, USA) and the results were expressed as mg of gallic acid equivalents (GAE) per L of beer. Data were expressed as the mean value for three measurements.

#### 2.7.2. Free-Radical-Scavenging Ability by the Use of a DPPH^•^ Radical Assay

The antiradical activity was determined using a DPPH^•^ radical assay [28]. Samples of beer or fruit juice (0.1 mL) were mixed with 2 mL of 0.04 mmol/L DPPH^•^ in ethanol and 0.4 mL of distilled water. After 10 min of incubation at room temperature, the absorbance was measured with a spectrophotometer at 517 nm using disposable polystyrene cuvettes. A calibration curve was prepared with Trolox solution (0.005 mmol/L). The data were expressed as Trolox equivalent (TE) of antioxidative capacity per 1 L of the beer (mmol TE/L). All measurements were performed in triplicate.

#### 2.7.3. Ferric Reducing/Antioxidant Power (FRAP) Assay

The FRAP assay is based on the reduction of ferric 2,4,6-tris(2-pyridyl)-1,3,5-triazine [Fe (III)-TPTZ] to the ferrous complex at low pH, followed by spectrophotometric analysis [29]. Briefly, the reagent was prepared by mixing 10 mmol 2,4,6-tris(2-pyridyl)-s-triazine (TPTZ)/L reagent with 20 mmol/L ferric chloride in acetate buffer (pH 3.6). Quantitative analyses were performed by the external standard method using ferrous sulphate (0.2 mmol/L) as the reference standard and correlating the absorbance (λ = 593 nm) with the concentration. 0.1 mL of beer/fruit juice sample was mixed in polystyrene cuvettes with 0.9 mL of distilled water and 3 mL of ferric complex. The data were expressed as Trolox equivalent (TE) of antioxidative capacity per 1 L of the beer (mmol TE/L). All measurements were performed in triplicate.

#### 2.7.4. Free-Radical-Scavenging Ability by the Use of an ABTS^+•^ Radical Cation

Another method used to measure the antioxidant activity of beers was the ABTS^+•^ radical cation assay [30]. Sample of beer or fruit juice (0.03 mL) were mixed with 3 mL of ABTS^+•^ solution with measured absorption of 0.700 at a wavelength of 734 nm. After 6 min the absorbance of samples was measured. Each sample was tested in triplicate. The data were expressed as mmol Trolox equivalent (TE) of antioxidative capacity per 1 L of the beer (mmol TE/L).

### 2.8. Adsorption of Volatile Compounds Using Solid Phase Microextraction (SPME)

Centrifuged and degassed beer (4 mL) was added to a 20 mL glass vial. 5 µL of 100 mg/100 mL aqueous emulsion internal standard solution (2-undecanone) was added. A magnetic stir bar was placed in the vial which was then closed with an aluminium membrane. The vial was placed on a IKA RCT basic magnetic stirrer (Guangzhou, China). The SPME holder needle with three-component universal fibre for SPME (DVB/CAR/PDMS 50/30 µm) was used to puncture the membrane. Heating to 40 °C and stirring was started and fibre was extended and held over the beer for 20 min. An analogical approach was used for each type of beer. The adsorption of volatile compounds contained in peeled hawthorn fruits added to beer was performed in order to find out which chemical compounds are characteristic for hawthorn fruit. Adsorption of compounds isolated from the hawthorn fruit was carried out in an analogous manner to the adsorption of volatile compounds from beer, but 5 g of hawthorn fruit instead of 4 mL of beer was added to the vial. The analysis of juice squeezed out of hawthorn fruit was carried out in an analogous manner to beer (4 mL juice volume). Measurements for each type of beer were performed in duplicate.

### 2.9. Gas Chromatography and Mass Spectrometry of Compounds Adsorbed on Fibre

Separation, identification and quantification of volatile compounds adsorbed on the fibre was carried out using a gas chromatograph connected to a Saturn 2000 MS Varian Chrompack mass detector (Palo Alto, CA, USA) with a ZB-5 column (Phenomenex, Torrance, CA, USA) (30 m length × 0.25 µm film thickness × 0.25 mm diameter).

Chromatographic conditions were carried out in accordance with the methodology of Calin-Sanchez et al. [31]. Scanning (1 scan/s) was carried out in the 35–400 m/z range using 70 mV electron ionization. The analyses were carried out using helium as a carrier gas with a flow rate of 1 mL/min using the following program for oven temperature: 40 °C at the beginning of the process, 5 °C/min up to 110 °C, 20 °C/min up to 270 °C. The initial temperature was maintained for 3 min. The injection port temperature of the chromatograph was 220 °C.

### 2.10. Sensory Analysis

The beers prepared in this study were subjected to an organoleptic assessment on a five-point scale using features such as clarity, aroma, colour, taste and overall impression. Beers were evaluated by a group of 15 people (21 to 32 years old), which consisted of 9 women and 6 men. Samples were given in coded plastic cups with a capacity of 200 mL. The temperature of the served beer was 8 °C.

### 2.11. Data Analysis

Volatile compounds separated from the beer were identified by comparing retention indexes (RI) with Kovats standards (KI exp. and KI lit.) and with NIST11 chemical standard libraries and by mass spectral analysis. Two standard matrices were also created (for the control sample chromatogram, i.e., beer without addition and hawthorn fruit). Chromatograms of the remaining samples (i.e., beer with addition of fruit and beer with addition of fruit juice) were integrated using retention time of compounds in those two standard matrices, using Mnova MS 12.0.1 software (Mestrelab Research, Santiago de Compostela, Spain). Data concerning content of volatile compounds in hawthorn juice and hawthorn fruit is available in Appendix A.

The results in this work were statistically analysed in the Statistica 12.5 program from Statsoft (Tulsa, OK, USA) using one-way ANOVA (α = 0.05). Differences between means were calculated using Duncan’s test (α < 0.05).

## 3. Results and Discussion

### 3.1. Concentration of Carbohydrates, Glycerol, Ethanol and pH Value of Tested Beers

The ethanol content in the tested beers ranged from 35.43–41.63 g/L (Table 1). The control beer without fruit addition (BC) had the highest content of ethanol, glycerol, maltose, maltotriose, as well as the highest pH value in relation to the other samples: beer with hawthorn fruit (BF) and beer with hawthorn juice (BJ). The addition of juice diluted and acidified the sample. This is indicated by a lower sugar content and lower pH characterising the samples with additives. BF compared to BJ, had a higher content of ethanol, glycerol, as well as maltose, maltotriose and dextrins.

The use of hawthorn fruit did not dilute the beer as much as juice addition. The fruit, after the primary fermentation process, was separated from the beer, so it did not significantly increase its final volume, in contrast to the samples with the addition of juice. This is reflected in the content of sugar, ethanol and glycerol per litre of beer, as well as in the pH value. Liu et al. analysed the sugar content in hawthorn fruits. The sugars identified in the fruits were: fructose, glucose, sucrose, sorbitol and myo-inositol [32]. The addition of hawthorn fruit or juice does not affect the content of maltose and maltotriose. The whole amount of these sugars was introduced into the beer with malt. The differences in sugar content between the samples are caused by the dilution of the samples by the addition of fruit or juice. Glucose content was not detected in any of the tested beers. Glucose is the sugar preferentially used by the yeast *Saccharomyces cerevisiae*. The final product usually does not contain this sugar or contains a small amount of it [33]. The addition of fruit or fruit juices reduces the pH of the finished product. Nardini and Garaguso showed that the pH value for the classic ale beer style ranges from 4.39 to 4.73, while for fruit beers the pH reached even around 3.5 [18]. This was also confirmed in the studies by Kawa-Rygielska et al., where the effect of adding Cornelian cherry varieties on beer parameters was analysed. Control beer, without the addition, had a pH value of 4.59, while beers with the addition of juice had a pH in the range of 3.43–3.71 [20]. The content and composition of sugars and acids in the product is an important factor shaping the quality of fruit, and thus also fruit products, by affecting the taste and reception of the product by consumers [34].

### 3.2. Concentration of Total Polyphenols and Antioxidative Activity

The use of hawthorn fruit or juice resulted in a significant increase in the polyphenol content as well as the antioxidant activity tested by FRAP, ABTS^+•^ and DPPH^•^ methods. The addition of fruit caused an increase in the content of polyphenolic compounds in beer by 1.4 times (279.6 mg GAE/L), while the use of fruit juice more than twice (410.0 mg GAE/L) (Table 2). A smaller increase in the polyphenol content was observed in BF compared to BJ. This can be explained by the poorer extraction of polyphenolic compounds from fruit into solution. Similar trends were observed for the beer’s antioxidant activity. The addition of juice increases the total antioxidant capacity (ABTS^+•^) 2.2 times, while the addition of fruit only 1.4 times. The ability to reduce the DPPH^•^ radical solution increased the in BJ up to 6 times while in BF the increase was much less noticeable (only about 1.2 times). The FRAP assay showed a 2.6 times increase in antioxidant activity for BJ, and 1.7 times BF compared to BC.

The addition of fruit or fruit juices in beer production is a method for significantly increasing the content of bioactive polyphenolic compounds in product. In the studies of Nardini and Garaguso, classic ale beers showed total polyphenol content in the range of 383–482 mg GAE/L, while beers with fruits (cherry, orange peel, grape, plum, raspberry, peach, apricot or apple) from 399 to up to 767 mg GAE/L [18]. The addition of fruits or fruit juices to beer can almost double the content of these biologically active compounds. BC contained 200.5 mg GAE/L polyphenolic compounds.

Fruit and vegetable peels can also be an interesting addition to the production of beer. Beers with eggplant skin extract were tested in a study by Horincar [22]. Polyphenol content in beers assessed in this study ranged from 0.443 to 0.610 mg GAE/mL, antioxidant capacity measured by DPPH^•^ assay was 1.287–1.306 mmol TE/mL, while ABTS^+•^ 0.095–0.107 mmol TE/mL. The content of polyphenols was of similar order as polyphenol content in BJ, but the antioxidant capacity of beer with addition of eggplant peel extract was significantly higher than antioxidant capacity of BJ. The main reason for such differences in tested beers was the amount of the addition used. The use of peels may be an interesting object for further research.

In addition to increasing antioxidant activity, polyphenolic compounds can affect the sensory characteristics of beer. They can introduce unpleasant bitter taste to beer and participate in the creation of turbidity [35]. Beer styles vary in polyphenol content. The type and dose of malts, hops, and the additions and technological processes used are factors which determine polyphenol content of the finished product. Habschied et al. [2] tested 12 commercially available beers including pilsner, lager, dark and black beers. The content of polyphenolic compounds ranged from 464.34 to 855.45 mg/L. Antioxidant activity of these beers was tested by DPPH^•^ assay and ranged from 0.4 to 0.61 mmol TE/100 mL. These values were higher than those obtained by fruit beers in our experiment. In a study by Kawa-Rygielska et al. [20] on beers with the addition of Cornelian cherry juice, it was shown that Cornelian cherry juice increased the content of polyphenols in a greater extent than the addition of hawthorn juice. However, another study on non-alcoholic beers with the addition of Cornelian cherry juice, carried out by Adamenko et al. [36] showed a smaller increase in the antioxidant activity and the content of polyphenols in beers than the increase in the antioxidant activity and content of polyphenols in beer with hawthorn juice, but greater than determined in beer with fruits.

### 3.3. Concentration of Volatile Compounds in Tested Beers

The main volatile chemicals in alcoholic beverages are ethanol and carbon dioxide. Other compounds, such as esters, play a crucial role in creating beer aroma, despite being present at very low concentrations [37]. We found almost no research about volatile compounds in beers with fruit addition [38] and no works in which authors compared differences in volatile composition of beer made with addition of fruit or juice. In BC, 45 volatile compounds were identified (Table 3). The largest groups among them were esters (20 compounds), sesquiterpenes (8 compounds), and alcohols (6 compounds). The greatest number (51) of volatile compounds was identified in BF. Like in the BC, the largest groups among the constituents were esters (25 compounds), sesquiterpenes (8 compounds) and alcohols (8 compounds). The analysis, in the case of BF, also revealed two peaks recognized as volatile constituents, which could not be identified (the mass spectra of unidentified constituents are available in the Appendix A). Other volatile compounds identified in BF, BJ and BC belonged to such groups as aldehydes, monoterpenes, ketones, organic acids and hydrocarbons. In a study by Gao et al. [39] about volatile compounds in hawthorn fruits, 61 volatile compounds were identified. Although more volatile constituents could be identified in fruit, our study managed to characterise more of the esters. The main reason for this phenomenon is probably yeast metabolism. Yeast cells can form esters by enzymatic chemical condensation of alcohols and organic acids [40]. The content of compounds identified in hawthorn fruit, such as ethyl butyrate, 1-hexanol, ethyl hexanoate, α-terpineol, ethyl octanoate, ethyl (4E)-4-decenoate and ethyl trans-2-decenoate was determined in all tested beers. The largest amount of compounds characteristic for hawthorn was recorded in BF and the smaller (except ethyl octanoate) in BJ. It is also worth noting that these compounds occur in a smaller amount, in the control sample. Therefore, they may also come from malt, hops or be products of yeast metabolism [41]. The addition of hawthorn fruits or hawthorn juice to beer introduced only six new compounds (acetic acid hexyl ester; propanoic acid 2-methyl, 2-methylbutyl ester; limonene, ethyl 5-methylhexanoate; octanoic acid methyl ester; isopentyl hexanoate). However, there were increases in the amount of almost all volatile compounds characteristic for the beer. The compound derived from hawthorn, which was found in the largest amount in BF, was α-terpineol, characterised by resinous and citrus aromas [42,43]. This constituent was found in the research about craft beers produced by mixed fermentation by *Torulaspora delbrueckii* non-conventional yeast and typical ale yeast [44]. In the control sample fermented solely with brewer’s yeast, α-terpineol was not detected. In the samples fermented with *Torulospora* culture, the amount of α-terpineol was 3 times higher than in BJ, but 4.5 times lower than in BF. As in the study by Canonico [44], the amount of α-terpineol in BC was minuscule (0.1019 µg/100 mL). Another compound, one of the esters characteristic to the hawthorn fruit, was ethyl hexanoate. Among the esters characteristic of hawthorn fruit aroma, the concentration of ethyl hexanoate in BF was the highest. It is a compound with a sweet, fruity aroma [45,46]. Its presence was demonstrated in all tested samples: BF (11.2222 µg/100 mL), BJ (1.3959 µg/100 mL) and BC (0.9681 µg/100 mL). It is a compound which was characterised in a study about dotted hawthorn by Kucharska et al. [46]. In dotted hawthorn, ethyl hexanoate constituted 26.5% of all volatile components, but in beer, ethyl hexanoate constituted only 3.9% of all volatiles. In a study about wines made from hawthorn fruit juice, conducted by Zhang et al. [47], it was shown that concentration of ethyl hexanoate in wines was 7 times higher than in BJ, 10 times higher than in BC, and similar to the concentration in BF. Another ester identified in hawthorn fruit and all tested beers was ethyl (4E)-4-decenoate, which is characterised by fruit (pineapple), cognac and wax flavours [48,49]. It was detected in the tested samples in the amount of 0.1267 µg/100 mL for BJ, 0.4505 µg/100 mL for BF and 0.0985 µg/100 mL for BC. Concentration of ethyl octanoate was highest in BJ (8.3716 µg/100 mL) and lowest in BC (0.0656 µg/100 mL). This compound is characterised by a fruity aroma and can originate from hawthorn fruit as well as yeast metabolism. Ethyl octanoate has been found in various beers analysed by other authors, such as in beer with sorghum addition made by Chenge et al., where its determined concentration was two times higher than in BJ [41,46]. A compound not belonging to the group of esters, which was identified in the fruits of hawthorn, was 1-hexanol, which is a compound from chemical family of alcohols. It has a sweet, alcoholic aroma [50,51]. Its highest concentration was found in BF (0.0819 µg/100 mL). BJ contained 0.0327 µg/100 mL of this compound, while BC contained 0.0164 µg/100 mL. This compound was also found in greater concentration in fruit beers with addition of quince than in control beer in study of Zapata et al. [38]. The most abundant sesquiterpenes in beers were humulene and humulene epoxide I. Concentration of humulene was the highest in BF (1.9659 µg/100 mL), and the lowest in BJ (0.4772 µg/100 mL). Meanwhile, humulene epoxide I was found in BF in the amount of 3.1127 µg/100 mL and BJ in the amount of 0.2812 µg/100 mL. In BC, the concentration of these compounds was greater than in BJ (respectively 0.5118 µg/100 mL and 0.2969 µg/100 mL). In classic beers, these compounds come from hops [52], while this study may indicate that these sesquiterpenes can be found in hawthorn fruit, but only in the flesh (because in BJ their quantity is smaller than in BC). Another compound, β-farnesene, with the lowest concentration in BJ, which is naturally found in classic beers, also comes from the sesquiterpene group [53]. It is a compound characterised by woody and citrus aromas [54,55]. Its concentration in BJ was 0.0136 µg/100 mL, while in BC it was 0.0492 µg/100 mL, and in BF 1.6383 µg/100 mL. The total concentration of all volatile compounds in the tested beers was the highest for BF which was 287.8529 µg/100 mL, which was 6.5 times higher than in BJ, which contained 43.8884 µg of volatile compounds per 100 mL of beer, and 8 times higher than BC, which contained 35.9348 µg of volatile compounds per 100 mL of beer.

### 3.4. Organoleptic Analysis

The aroma, colour, clarity, foaminess and overall impression of beer were assessed. The beer that achieved the highest rating in terms of aroma, taste, clarity and overall impression was BF (Table 4, Figure 1). This is in line with the results of research carried out by Adamenko et al. [36] on beers with the addition of Cornelian cherry, where beers with the addition of fruit also received better marks in terms of taste than control beers. Statistical analysis showed no significant differences between BC and BF clarity, although the addition of juice negatively affected this attribute. The use of whole fruit caused a significant deterioration in beer foaminess. Control beer got the highest grade for foaminess; however, statistical analysis did not show a significant difference between the BC and BJ results. The addition of juice does not worsen the beer foam quality. It is possible that a high concentration of polyphenols in beer can have a detrimental effect on beer foaming [2]. Although some authors show a positive effect of high polyphenols on beer foam stability [56]. The results obtained are contradictory. It has not been clearly defined how polyphenolic compounds affect beer foam. BF achieved the best note for taste criterion. In this aspect of sensory analysis, it obtained significantly higher results than other analysed beers. BJ and BF beers received better aroma ratings than BC. This may be due to a higher concentration of esters characterised by floral and fruity notes (ethyl hexanoate or ethyl octanoate). Zapata et al. [38] presented similar results. They analysed the composition of the quince beer aroma. The study showed that the higher concentration of ethyl esters resulted in better scores in sensory analysis.

## 4. Conclusions

The study indicates that the addition of hawthorn (*Crataegus punctata*) makes it possible to obtain beer with an increased total content of polyphenolic compounds and antioxidant activity. Beer without hawthorn addition had the highest content of ethanol carbohydrates, glycerol and highest pH value. The total content of polyphenolic compounds was affected by both the addition of fruit and the form of the fruit added. Hawthorn in the form of juice made it possible to obtain a much higher content of bioactive polyphenolic compounds in beer than addition of fruit. Nevertheless, beer with fruit addition was characterised by the greatest amount of volatile compounds, eight times higher than in the control sample. Enrichment of the beer by hawthorn fruit mostly increased the concentration of the volatile compounds characterised by fruity and sweet aromas. The results of sensory analysis indicate that addition of hawthorn fruit results in an improvement of such characteristics as taste, aroma, clarity and overall impression in a higher degree than hawthorn juice addition. Hawthorn fruit and its juice can be used as complementary raw material in the production of beer to increase its biological activity and improve its taste and aroma. It can also contribute to greater consumer interest in the product.

## Figures and Tables

**Figure 1 foods-09-00775-f001:**
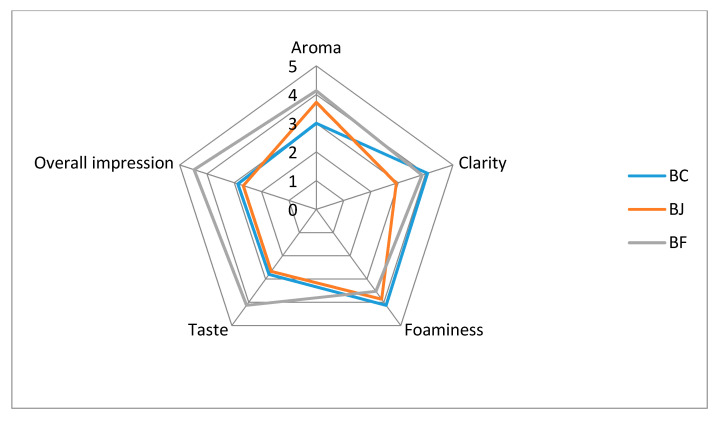
Organoleptic analysis of tested beers.

**Table 1 foods-09-00775-t001:** Concentration of carbohydrates, glycerol, ethanol and pH value in tested beers.

Beer Type ^1^	Ethanol	Glycerol	Maltose	Maltotriose	Dextrin	Glucose	pH
	[g/L]	[g/L]	[g/L]	[g/L]	[g/L]	[g/L]	
BC	41.63 ± 0.02 ^a^	2.23 ± 0.01 ^a^	1.05 ± 0.002 ^a^	1.13 ± 0.003 ^a^	17.58 ± 0.01 ^a^	n.d.	3.87 ± 0.02 ^a^
BJ	35.42 ± 0.2 ^c^	1.95 ± 0.01 ^c^	0.8 ± 0.015 ^c^	0.92 ± 0.007 ^c^	14.21 ± 0.05 ^c^	n.d.	3.5 ± 0.02 ^c^
BF	37.23 ± 0.03 ^b^	2.07 ± 0.04 ^b^	0.9 ± 0.001 ^b^	0.98 ± 0.015 ^b^	15.77 ± 0.3 ^b^	n.d.	3.7 ± 0.01 ^b^

^1^ BC—control beer; BJ—beer with hawthorn juice; BF—beer with hawthorn fruit, Values are expressed as mean (*n* = 3) ± standard deviation. Mean values with different letters (a, b, c) within the same column are statistically different (*p*-value < 0.05).

**Table 2 foods-09-00775-t002:** Antioxidant activity and concentration of polyphenols in tested beers and juice pressed from the hawthorn fruit.

Analysed Sample ^1^	Polyphenol Concentration	ABTS^+•^	DPPH^•^	FRAP
	mg GAE/L	mmol TE/L	mmol TE/L	mmol TE/L
BC	200.5 ± 1.9 ^d^	0.936 ± 0.09 ^d^	0.352 ± 0.03 ^c^	0.512 ± 0.01 ^d^
BJ	410.1 ± 11.8 ^b^	2.041 ± 0.12 ^b^	2.175 ± 0.01 ^b^	1.35 ± 0.02 ^b^
BF	279.6 ± 2 ^c^	1.356 ± 0.11 ^c^	0.443 ± 0.04 ^c^	0.869 ± 0.01 ^c^
J	2633.9 ± 27.9 ^a^	11.03 ± 0.32 ^a^	17.22 ± 0.29 ^a^	8.52 ± 0.13 ^a^

^1^ BC—control beer; BJ—beer with hawthorn juice; BF—beer with hawthorn fruit, J—hawthorn juice. Values are expressed as mean (*n* = 3) ± standard deviation. Mean values with different letters (a, b, c, d) within the same column are statistically different (*p*-value < 0.05).

**Table 3 foods-09-00775-t003:** Content of volatile compounds in tested beers.

				Kovats Indices			Concentration in Beer µg/100 ML ^1^		
	Compound Name	Retention Time	Chemical Family	KI Exp.	KI NIST	CAS Number	BJ	BF	BC
1	1-Butanol, 3-methyl-	4.208	Alcohols	724	735	123-51-3	5.7104 ± 2.8229 c	11.9594 ± 4.9811 b	18.4103 ± 9.6497 a
2	Isobutyl acetate	5.145	Esters	763	771	110-19-0	0.0257 ± 0.0133 b	0.2048 ± 0.0954 a	0.0328 ± 0.0147 b
3	Butanoic acid. ethyl ester	5.852	Esters	799	802	105-54-4	0.0812 ± 0.0622 b	0.5324 ± 0.3472 a	0.0492 ± 0.0251 b
4	1-Hexanol	7.899	Alcohols	872	868	111-27-3	0.0327 ± 0.0142 b	0.0819 ± 0.0533 a	0.0164 ± 0.0086 c
5	1-Butanol, 3-methyl-, acetate	8.095	Esters	872	876	123-92-2	3.0275 ± 1.3892 b	18.4305 ± 6.4922 a	1.6409 ± 0.7294 b
6	5-Hepten-2-one, 6-methyl-	11.656	Ketones	987	986	110-93-0	0.0139 ± 0.0068 b	0.0819 ± 0.0412 a	0.0153 ± 0.0089 b
7	β-Myrcene	11.782	Monoterpenes	994	991	123-35-3	0.0137 ± 0.0101 b	0.1229 ± 0.0844 a	0.0164 ± 0.0114 b
8	Hexanoic acid, ethyl ester	12.034	Esters	1001	1000	123-66-0	1.3959 ± 0.9982 b	11.2221 ± 6.4921 a	0.9681 ± 0.06742 b
9	Acetic acid, hexyl ester	12.508	Esters	1017	1011	123-35-3	0.0059 ± 0.0036 b	0.1229 ± 0.0712 a	Trace
10	Propanoic acid, 2-methyl-, 2-methylbutyl ester	12.577	Esters	1019	1016	2445-69-4	0.0042 ± 0.0021 b	0.6963 ± 0.3152 a	Trace
11	p-Cymene	12.83	Aromatic hydrocarbons	1028	1025	99-87-6	0.0079 ± 0.0023 c	0.6144 ± 0.2197 a	0.0328 ± 0.0094 b
12	Limonene	12.955	Monoterpenes	1032	1030	5989-54-8	0.0218 ± 0.0074 b	0.1638 ± 0.0852 a	Trace
13	Butanoic acid, 3-methylbutyl ester	13.937	Esters	1062	1056	106-27-4	0.0041 ± 0.0019 c	0.1229 ± 0.0622 a	0.0164 ± 0.0072 b
14	Ethyl 5-methylhexanoate	14.035	Esters	1066	1072	10236-10-9	0.0119 ± 0.0041 a	0.0819 ± 0.0332 a	Trace
15	1-Octanol	14.316	Alcohols	1075	1071	111-87-5	0.0972 ± 0.0386 c	7.4541 ± 2.8264 a	0.5743 ± 0.2848 b
16	2-Nonanone	14.96	Ketones	1096	1092	821-55-6	0.0119 ± 0.0049 c	41.5231 ± 11.9227 a	2.7894 ± 1.0048 b
17	Linalool	15.2	Alcohols	1099	1099	78-70-6	0.8554 ± 0.2652 b	1.1468 ± 0.4392 a	0.0824 ± 0.0312 c
18	2-Nonen-1-ol	15.311	Alcohols	1099	1105	22104-79-6	0.1584 ± 0.0466 c	43.4859 ± 13.5984 a	2.7894 ± 0.9982 b
19	Phenylethyl Alcohol	15.607	Esters	1114	1116	1960-12-08	3.0156 ± 0.9954 a	1.1877 ± 0.05572 b	0.0492 ± 0.0235 c
20	Octanoic acid, methyl ester	15.945	Esters	1127	1126	106-32-1	0.0396 ± 0.0178 b	0.2431 ± 0.0861 a	trace
21	Endo-Borneol *	17.199	Monoterpenes	1177	1167	464-43-7	0.0158 ± 0.0084 c	0.2457 ± 0.0973 a	0.0656 ± 0.0221 b
22	Benzoic acid, ethyl ester	17.311	Esters	1180	1171	93-89-0	0.0976 ± 0.0394 c	10.1163 ± 3.8572 a	0.7548 ± 0.2362 b
23	Octanoic acid	17.674	Organic acids	1192	1180	124-07-2	1.3425 ± 0.5283 a	0.5734 ± 0.2271 b	0.0492 ± 0.0198 c
24	α-Terpineol	17.772	Monoterpene	1194	1190	98-55-5	5.7758 ± 2.2853 b	73.1488 ± 23.0739 a	0.1019 ± 0.0476 c
25	Octanoic acid, ethyl ester	17.858	Esters	1196	1196	106-32-1	8.3716 ± 2.3497 a	0.2457 ± 0.0612 b	0.0656 ± 0.0227 c
26	Decanal	18.028	Aldehydes	1207	1206	112-31-2	0.1465 ± 0.0381 b	1.2287 ± 0.4776 a	0.0164 ± 0.0048 c
27	Acetic acid, octyl ester	18.126	Esters	1217	1211	112-14-1	0.0178 ± 0.0061 c	4.0547 ± 1.1286 a	0.3774 ± 0.0982 b
28	Citronellol	18.46	Alcohols	1238	1229	106-22-9	0.5307 ± 0.2163 a	Tr ace	Trace
29	Benzeneacetic acid, ethyl ester	18.709	Esters	1254	1246	101-97-3	0.0139 ± 0.0047 c	20.3555 ± 4.3992 a	2.0182 ± 0.8642 b
30	Isopentyl hexanoate	18.752	Esters	1256	1252	2198-61-0	0.0079 ± 0.0032 b	0.0287 ± 0.0092 a	trace
31	Acetic acid, 2-phenylethyl ester	18.879	Esters	1263	1258	103-45-7	2.9661 ± 1.0873 a	0.3277 ± 0.0932 b	0.0164 ± 0.0052 c
32	1-Decanol	19.077	Alcohols	1277	1273	112-31-2	0.0812 ± 0.0272 a	0.0416 ± 0.0138 b	0.0328 ± 0.0098 b
33	2,4-Heptadienoic acid, 6-methyl-, ethyl ester	19.146	Esters	1282	1293	10236-06-03	0.1327 ± 0.4921 b	0.2457 ± 0.0733 a	0.0164 ± 0.0043 b
34	Methyl geranate	19.728	Esters	1339	1326	214-712-6	0.4396 ± 0.0963 b	1.4744 ± 0.3896 a	0.3282 ± 0.1158 b
35	Octanoic acid, 2-methylpropyl ester	19.938	Esters	1352	1348	5461-06-03	0.2198 ± 0.0624 b	0.3686 ± 0.1374 a	0.1641 ± 0.0556 b
36	Citronellol acetate	20.023	Esters	1357	1354	150-84-5	0.0973 ± 0.0372 b	0.3277 ± 0.0992 a	0.0985 ± 0.0313 b
37	Unknown sesquiterpene	20.149	Sesquiterpenes	1369			1.8197 ± 0.6294 a	0.6553 ± 0.3278c	1.2799 ± 0.3775 b
38	Ethyl (4E)-4-decenoate	20.291	Esters	1384	1377	76649-16-6	0.1267 ± 0.0352 b	0.4505 ± 0.0966 a	0.0985 ± 0.0217 b
39	Ethyl trans-2-decenoate	20.347	Esters	1389	1389	7367-88-6	0.7702 ± 0.2432 b	7.5366 ± 2.8836 a	0.5087 ± 0.1764 b
40	Decanoic acid, ethyl ester	20.418	Esters	1396	1396	110-38-3	4.1086 ± 1.2883 a	1.6792 ± 0.4732 b	0.5127 ± 0.2296 c
41	Dodecanal	20.561	Aldehydes	1412	1409	112-54-9	0.0396 ± 0.0084 b	0.3277 ± 0.0912 a	0.0492 ± 0.0068 b
42	β-Caryophyllene	20.786	Sesquiterpenes	1440	1419	87-44-5	0.0772 ± 0.0252 b	0.3686 ± 0.0992 a	0.0476 ± 0.0118 b
43	Octanoic acid, 3-methylbutyl ester	20.87	Esters	1453	1446	2035-99-6	0.1148 ± 0.0326 b	1.4335 ± 0.3774 a	0.0985 ± 0.0264 b
44	(E)-β-Famesene	20.984	Sesquiterpenes	1464	1457	18794-84-8	0.0436 ± 0.0092 b	1.6383 ± 0.4694 a	0.0492 ± 0.0111 b
45	Humulene	21.069	Sesquiterpenes	1474	1454	6753-98-6	0.4772 ± 0.0958 b	1.9659 ± 0.5312 a	0.5118 ± 0.1322 b
46	α-Muurolene	21.224	Sesquiterpenes	1493	1485	10208-80-7	0.0178 ± 0.0054 b	0.2457 ± 0.0618 a	0.0164 ± 0.0051 b
47	Pentadecane	21.296	Hydrocarbons	1500	1500	629-62-9	0.0139 ± 0.0058 c	0.3686 ± 0.1372 a	0.0492 ± 0.0126 b
48	δ-Cadinene	21.561	Sesquiterpenes	1538	1524	483-76-1	0.1267 ± 0.0344 b	0.9838 ± 0.2258 a	0.1149 ± 0.0376 b
49	Unknown compound	21.687		1557			0.0317 ± 0.0076 a	0.0023 ± 0.0017 b	Trace
50	β-Calacorene	21.744	Sesquiterpenes	1565	1563	50277-34-4	0.0475 ± 0.0122 b	0.0819 ± 0.0315 a	0.0328 ± 0.0087 b
51	Nerolidol	21.786	Sesquiterpenes	1574	1565	7212-44-4	0.0297± 0.0102 b	0.4505 ± 0.0936 a	0.0245 ± 0.0097 b
52	Dodecanoic acid, ethyl ester	21.955	Esters	1590	1595	106-33-2	0.9722 ± 0.2318 b	14.2939 ± 3.5972 a	0.6563 ± 0.2532 b
53	Humulene epoxide I	22.167	Sesquiterpenes	1607	1604	19888-33-6	0.2812 ± 0.0676 b	3.1127 ± 0.9136 a	0.2969 ± 0.0754 b
	Total						43.8884	287.8529	35.9348

^1^ BC—control beer; BJ—beer with hawthorn juice; BF—beer with hawthorn fruit. Values are expressed as mean (*n* = 2) ± standard deviation. Mean values with different letters (a, b, c) within the same line are statistically different (*p*-value < 0.05).

**Table 4 foods-09-00775-t004:** Organoleptic analysis of tested beers.

Beer Type ^1^	Aroma	Clarity	Foaminess	Taste	Overall Impression
BC	3.02 ± 0.2 ^c^	4.07 ± 0.21 ^a^	4.13 ± 0.19 ^a^	2.81 ± 0.17 ^b^	2.87 ± 0.19 ^b^
BJ	3.73 ± 0.15 ^b^	2.93 ± 0.18 ^b^	3.87 ± 0.19 ^ab^	2.67 ± 0.21 ^b^	2.67 ± 0.19 ^b^
BF	4.13 ± 0.19 ^a^	3.87 ± 0.24 ^a^	3.53 ± 0.13 ^b^	4.13 ± 0.19 ^a^	4.47 ± 0.17 ^a^

^1^ BC—control beer; BJ—beer with hawthorn juice; BF—beer with hawthorn fruit. Values are expressed as mean (n = 15) ± standard deviation. Mean values with different letters (a, b, c) within the same column are statistically different (*p*-value < 0.05).

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
