# Peer review of "Assessment of Volatiles and Polyphenol Content, Physicochemical Parameters and Antioxidant Activity in Beers with Dotted Hawthorn (Crataegus punctata)"

_foods, 2020, doi:10.3390/foods9060775_

Round 1
Reviewer 1 Report
The introduction is long and sometimes refers aspects not explored in the experimental part, such as the extraction of oils, but in experimental part it is mentioned the juice addition, not oils addition.
The introdution begin with reference 19 and throughout the text there is never an order, which does not facilitate the final bibliography writing. I do not find 18 reference in the text.
More than 77 % of the bibliography cited have more than 5 years old and craft beer is very recent!

Reviewer 2 Report
The author proposed an interesting work on the exploitation of dotted hawthorn to improve quality and functional properties of beer. However, two rounds of revisions are needed, in my opinion, to reach the quality of the journal.
-) Please format the manuscript following the journal template (in particular for the tables/figure).
-) Please revise the introduction to clarify the gap of knowledge you aim to cover. In this issue, clarity and adequate references are crucial. E.g. line 61: some studies (ref?); Line 63: almost? What is clear to me is that "No work has been found in which it was analysed how the addition of hawthorn fruit to beer modifies the content of volatile compounds, the antioxidant activity and the content of polyphenols in beers". But in terms of the use of fruits and beer to modulate volatile content, polyphenols content and antioxidant activity, what new you aim to demonstrate. Please clarify.
-) Please provide the outputs of statistical analysis also for Table 3.
-) The results are poorly discussed. Please improve (also in the light of the improved introduction).
-) The general impression is to find a manuscript written fastly. Please use this round of revision also to improve the overall quality of writing/presentation.
Round 2
Reviewer 1 Report
The work has been improved, the introduction is more objective and more recent bibliography is referred. Also the references are ordered.
Only two spaces were missing on line 22 and 23:
where is 100mL of beer, must be 100 mL of beer
Author Response
Dear Reviewer.
Thank you for your insight.
We cleared the double spaces in the introduction and added an interval before each "mL".
We are glad to receive positive review.
We wish you all the best.
Reviewer 2 Report
Dear authors, thank you for your revisions. Please do your best to improve the general QUALITY and the HARMONISATION of the manuscript. My suggestions are only examples to demonstrate you that the manuscript can be generally improve. It is a way to suggest a direction to integrate the WHOLE manuscript. Please do your best with a further critical reading. My impression is that in the whole you did a good experimental plan with a poor effort in presentation/valorization. Please take the time to do a good job. It is in our global interest.
-) I found some ‘jumps’ in the lines numbers.
-) I suggest you to include in the title the polyphenols…e.g. “Assessment of volatiles and polyphenols content, physicochemical parameters and antioxidant activity”
-) You article aim to evaluate the impact of dotted hawthorn on beer volatiles, physicochemical parameters, polyphenols, and antioxidant activity. I was expecting a balanced introduction. The part of polyphenols is well articulated. On the opposite, the relevance of volatiles and of antioxidant activity were poorly introduced. Please add a paragraph on the relevance of volatiles (different cereal, https://www.mdpi.com/2311-5637/6/2/53; different hop, https://www.sciencedirect.com/science/article/abs/pii/S0963996918304241; different fermentation, https://www.mdpi.com/2306-5710/3/4/51; FRUIT…). Also improve the part on antioxidant (https://www.mdpi.com/1420-3049/25/3/731/htm, doi:10.3390/molecules25112582, doi:10.3390/molecules24081568, https://www.mdpi.com/2304-8158/9/2/238/htm, https://www.mdpi.com/2218-273X/10/3/400)
-) In your document, you used both North American and British/Australian spellings. Please uniform (my suggestion is to follow only the British one.
-) A suggestion for your future studies. If you have the control beer for juice addition you have to add a quantity of water/saline solution/sugar solution equal to the quantity of juice you added. Otherwise, you have a dilution effect that can impact on the ‘life’ of yeast…are you observing the impact of the dilution on the yeast-based bioprocess or the addition of juice? Impossible to be sure about this…in particular for volatile compound. I cannot ask to perform new experiments. I only ask you to discuss this point among the discussion concerning the volatiles…underlining that it is reasonable to address the observed changes to juice addition…
-) The sensory evaluation is mentioned in the abstract but not in the title and in the introduction. Please harmonise as much as possible the manuscript. Additionally, it is crucial underline the novelty of your study…how many studies tested at the same time polyphen, antiox, volatiles, sensory attributes for fruit beer? Maybe it is interesting to underline this point. (Please consider that all these changes in the introduction have not to increase the confusion. The introduction has to be clear and flows following a unique ‘fil rouge’…
-) Have the comparison ‘fruit versus fruit juice’ been reported in previous studies? This is one further point that can be clarified and better presented throughout the whole manuscript. I was expecting to find a sort of SWOT analysis on the use of ‘fruit versus fruit juice’, also in the light of previous literature, considering all the aspects you tested.
-) The section 3.4 is poorly commented. Please improve highlighting the connection with the findings of the other sections and with literature in the field.
-) The section on the volatile compounds is well described in terms of differences and associated notes. But what about the literature about the main volatiles you are able to improve (in comparison with other beer with fruit addition)?
-) Please check the following points:
Line 19: solid-phase
Line 39: beer, including
Line 50-51: disease… such as …antioxidant…please verify…
Line 52: in my opinion, when we claim positive effects for moderate alcoholic beverage consumption, we have always to remember that ethanol is a toxic molecule for human health also in low quantity.
Line 94: In the literature,
Line 94: , some studies are treating about…please cite these studies
Line 95: only a few studies…please cite these studies
Line 105: please verify if the fruit addition have to be considered as an additive or an ingredient in beer production. Please use the term additive in the sense of additive used in food processing https://ec.europa.eu/food/safety/food_improvement_agents/additives/database_en or clarify. Please check this aspect in the whole manuscript.
Line 106: The study aimed to determine
Line 282: “The use of hawthorn fruit did not dilute the beer”. Lines 289-291: “The differences in sugars content between the samples are caused by the dilution of the samples by the addition of fruit or juice”. Is there a dilution effect in the case of fruit???
Line 306: in the final products.
